# Neutralizing Antibodies: Role in Immune Response and Viral Vector Based Gene Therapy

**DOI:** 10.3390/ijms26115224

**Published:** 2025-05-29

**Authors:** Tatiana S. Tsaregorodtseva, Aigul A. Gubaidullina, Beata R. Kayumova, Alisa A. Shaimardanova, Shaza S. Issa, Valeriya V. Solovyeva, Albert A. Sufianov, Galina Z. Sufianova, Albert A. Rizvanov

**Affiliations:** 1Institute of Fundamental Medicine and Biology, Kazan Federal University, 420008 Kazan, Russia; tascaregorodceva@kpfu.ru (T.S.T.); aygagubaydullina@kpfu.ru (A.A.G.); berkayumova@kpfu.ru (B.R.K.); alisashajmardanova@kpfu.ru (A.A.S.); vavsoloveva@kpfu.ru (V.V.S.); 2Department of Genetics and Biotechnology, St. Petersburg State University, 199034 St. Petersburg, Russia; shaza.issa98@outlook.com; 3Department of Neurosurgery, Sechenov First Moscow State Medical University of the Ministry of Health of the Russian Federation (Sechenov University), 119991 Moscow, Russia; sufianov_a_a@staff.sechenov.ru; 4The Research and Educational Institute of Neurosurgery, Peoples’ Friendship University of Russia (RUDN), 117198 Moscow, Russia; 5Department of Pharmacology, Tyumen State Medical University, 625023 Tyumen, Russia; sufarm@mail.ru; 6Division of Medical and Biological Sciences, Tatarstan Academy of Sciences, 420111 Kazan, Russia

**Keywords:** antibodies, neutralizing antibodies, gene therapy, immune response, humoral immunity, viral vectors, overcoming the immune response, tolerance to gene therapy, immunogenicity of vectors

## Abstract

Neutralizing antibodies (nAbs) are an important component of the immune system, which plays a dual role in modern medicine. On the one hand, they significantly limit the effectiveness of gene therapy based on viral vectors, reducing the effectiveness of treatment of diseases such as spinal muscular atrophy, which is especially evident with repeated administration of therapeutic vectors. On the other hand, nAbs is a promising tool for combating viral infections. This review systematizes current data on the mechanisms of nAbs formation against AAV vectors, analyzes the factors influencing their production, and discusses strategies to overcome this limitation, including modification of vectors and the development of methods to suppress the immune response. Special attention is paid to the prospects of using nAbs as therapeutic agents against viral infections. The key problems and possible directions of research development in this area are considered, which is important for improving approaches to the treatment of both rare genetic and infectious diseases.

## 1. Introduction

Thanks to the rapid advancement of biomedical technologies, it has now become possible to develop new high-tech treatment methods for rare (orphan) diseases, which affect more than 350 million people worldwide [1]. Despite scientific breakthroughs in gene and cell therapy, more than 90% of rare diseases still lack effective treatment options [2].

The development of genetic engineering and medical genetics has enabled the creation of effective gene therapy approaches for various diseases based on adeno-associated viruses (AAV). AAV vectors are successfully used for the treatment of several rare genetic disorders. For instance, the gene therapy drug “Zolgensma”, which is based on AAV9 (adeno-associated virus serotype 9) vector, has been approved by the U.S. Food and Drug Administration (FDA) and is effectively used for treating children with spinal muscular atrophy (SMA). However, studies have shown that the therapy does not yield the expected effect in some patients, while others exhibit rapid and significant improvement [3].

A long-term therapeutic effect of gene therapy can be achieved through repeated administration of viral vectors [4,5]. However, the efficacy of repeated administration may be reduced since the patient’s immune system, after the initial vector injection, begins producing neutralizing antibodies (nAbs), which decrease the effectiveness of following viral transduction [6,7].

## 2. Definition of nAbs

Antibodies are antigen-binding proteins found both on the surface of B cells, and in secreted forms produced by plasma cells. They circulate in the bloodstream, and contribute to humoral immunity [8]. Although all antibodies share common structural features and execute the same effector functions, their diversity results from specificity for different antigens [9].

Structurally, these molecules are Y-shaped heterodimers composed of two light polypeptide chains and two heavy chains. Both light chains have a mass of about 25 kDa, while the minimum mass of the heavy chains is 50 kDa, which varies depending on the immunoglobulin isotype [10]. The heavy and light chains are held together by multiple disulfide bridges and non-covalent interactions, with the number of such connections also depending on the immunoglobulin isotype [11].

Antibodies can be classified based on their functional components. Two antigen-binding fragments, known as F_ab_, are responsible for pathogen binding and neutralization [12]. These fragments are linked to the crystallizable fragment, F_c_, through a flexible hinge region, providing F_ab_ with significant conformational flexibility relative to the F_c_-domain. This structural feature enables F_ab_ to interact with antigens regardless of their spatial orientation [13].

Antibodies are categorized into isotypes based on specific gene segments—alpha, mu, gamma, epsilon, or delta—that associate with the variable region [14]. In humans, antibody subclasses are encoded by the following genes: two alpha variants (IgA1 and IgA2), four gamma variants (IgG1, IgG2, IgG3 and IgG4), one mu variant (IgM), one epsilon variant (IgE) and one delta variant (IgD). Each subclass performs specific functions to eliminate specific pathogen types [10]. Effective neutralization is primarily mediated by IgA, IgM, and IgG [15]. For instance, in humans, AAV administration rapidly induces a significant increase in IgM levels, followed by a rise in IgG and IgA, which show high neutralizing titers following intramuscular or intravenous injection. The main neutralizing antibody is IgG [16].

Neutralization is defined mainly in two ways: first, as the “loss of infectivity occurring when an antibody molecule(s) binds to a viral particle, typically without requiring any additional agents” [17], and the second, as “reducing viral infectivity by binding antibodies to the surface of viral particles (virions), thereby blocking the stage of the viral replication cycle that precedes transcription or synthesis encoded by the virus” [18]. In enveloped viruses, this blockade occurs before cellular entry, while in non-enveloped viruses, it may occur post-entry [19].

The protective effect provided by nAbs is not limited to neutralizing free viral particles but also includes multiple mechanisms targeting infected cells [20]. In addition, the size of nAbs is important, which is comparable to the size of the classic viral envelope spike [21].

In general, nAbs, capable of recognizing nearly all AAV serotypes, were identified in most subjects. This phenomenon may result from multiple infections caused by exposure to different wild-type AAV serotypes, as well as the broad cross-reactivity among antibodies. Cross-reactivity is most likely due to the high amino-acid sequence homology observed between the capsids of different AAV serotypes [22]. Studies show that even minor structural modifications in the viral capsid can boost the immune response, leading to the production of antibodies that effectively neutralize multiple serotypes [23].

To avoid confusion, it should be noted that there are non-neutralizing antibodies (nnAbs), that also have neutralizing activity, but they do not block the virus, and are less effective than nAbs [17,21]. There are also the broadly neutralizing antibodies (bnAbs), that target pathogens capable of rapidly altering their antigenic profiles through mutation and genetic recombination. Examples of such pathogens include human immunodeficiency virus (HIV), influenza viruses, hepatitis C virus, *Streptococcus pneumoniae*, the malaria-causing parasite *Plasmodium falciparum*, and *Trypanosoma brucei*, which causes sleeping sickness [19]. Moreover, certain antibodies have been reported to cross-neutralize SARS-CoV-2 and other coronaviruses [21,24,25].

### 2.1. Mechanisms of Action of nAbs

The mechanisms of neutralization have been extensively studied over many years. Today, it is becoming more evident that the neutralization process mediated by antibodies in vitro occurs using a wide range of different mechanisms, highlighting the complex nature of antibody–antigen interactions (Figure 1) [18]. Here, we consider the primary mechanisms of virus neutralization, particularly for enveloped viruses. The basic principle of these mechanisms is the binding to functional molecules responsible for viral entry, which are typically located on the surface of infectious virions, often in the form of envelope glycoprotein spikes [26].

Nevertheless, it should be noted that significant differences in scientific opinions persist in this area, and different viruses may use diverse neutralization mechanisms depending on the specific conditions [18].

#### 2.1.1. Disruption or Conformational Modifications of Viral Spikes

This mechanism prevents virions from attaching to host cell receptors. According to this model, neutralization occurs when an antibody occupies a substantial proportion of accessible epitopes on the virion surface, leading to the inhibition of viral adhesion to host cells or impairing the entry (fusion) process. A key requirement for the “occupancy” or “coating” model is that the neutralizing activity of antibodies must correlate with their affinity for antigens located on the virion surface [27].

Structural studies have collectively demonstrated that antibodies employ diverse strategies to effectively recognize a wide range of antigenic surfaces and epitope localizations, identifying vulnerable sites in the protective viral envelope [28,29]. Practically no region on the surface of viral glycoproteins could be entirely insusceptible to adaptive immune responses. Notably, many Env molecules exhibit metastable properties, existing in a pre-fusion conformation within the virion structure [30]. Interaction with host cell receptors triggers substantial structural changes that facilitate viral entry into cells. Several cases have been documented in which nAbs bind to recombinant Env proteins, inducing conformational changes that render them nonfunctional and potentially irreversibly block viral entry [27].

#### 2.1.2. Aggregation by nAbs

Antibody-induced virion aggregation is considered a phenomenon distinct from direct neutralization. However, since this mechanism reduces the infectivity of viral particles, it can be classified as a specific neutralizing effect, although it shows complex interactions between both antibody and virion concentrations [31]. In general, the dynamics of virion aggregation relative to antibody concentration follow a bell-shaped curve: at low antibody concentrations, active cross-linking of antigens on individual virions promotes aggregate formation, while at higher antibody levels, virions bind to one another via bridge formation. However, when epitopes on the virion surface become saturated, further cross-linking is no longer possible [32].

In an in vivo context, the aggregation effect can vary significantly depending on the availability of target cells, as well as the characteristics of antibodies and virions in the surrounding environment. Interestingly, phagocytosis of formed aggregates by immune cells may not only contribute to direct neutralization but also facilitate additional inhibition through mechanisms that do not necessarily involve direct neutralization itself [33]. This highlights the complexity of antibody–virus interactions, leading to a diversity of immune responses [17,32].

#### 2.1.3. Steric Obstruction Following Viral Attachment

As previously mentioned, antibodies are relatively large molecules, with dimensions comparable to those of a typical viral spike protein. This characteristic plays a crucial role in the mechanism of neutralization following viral attachment. The fusion of viral and host cell membranes, a crucial step for viral entry, is a complex process. It can be blocked if a bulky antibody molecule interacts with the relevant site on the virus [17].

A case has been described where antibodies block West Nile virus infection at a post-entry stage by isolating the virus within acidic compartments and preventing its release into the cytoplasm. After this, the antibodies help eliminate the virus from the body [34].

#### 2.1.4. Intracytoplasmic Neutralization

For decades, it was widely believed that antibodies primarily provide immune protection in extracellular environments. However, growing evidence increasingly shows that antibodies can function effectively inside cells as well [32]. Intracellularly active antibodies target a broad spectrum of viral proteins, including both those expressed on the surface of viral particles and internal proteins found exclusively in the cytosol. This specificity suggests that viral neutralization can occur at various stages of the viral life cycle [35].

Antibodies have the ability to neutralize viral infections within endosomes, thereby blocking the release of viral genetic material. Additionally, antibodies can interfere with molecular mechanisms associated with viral particle fusion with the cell membrane, and the release of newly formed virions. Intracellular antibody activity also includes interactions with receptors such as TRIM21 (Tripartite motif-containing protein 21), an intracellular antibody effector involved in proteolytic pathways, playing a key role in the recognition and degradation of viral agents in the cytosol [36,37].

#### 2.1.5. Steric Blockade of the Receptor-Binding Site Before Virus Attachment

The basic idea of steric blockade is that the neutralizing antibody physically prevents the virus from interacting with its receptor on the surface of the host cell. This is due to the fact that the antibody binds to a special viral protein that is directly involved in binding to the receptor [38]. The higher the affinity of nAb to the viral protein, the more effectively it will compete with the receptor for binding [39].

#### 2.1.6. Prevention of Conformational Changes Necessary for the Fusion of Virus and Cell Membrane

After attachment, the virus must fuse with the cell membrane to deliver its genetic material inside the cell, however, using this mechanism, nAbs can bind to viral proteins involved in fusion and block the conformational changes necessary for this process. Using the example of SARS-CoV-2, its spike protein S consists of an N-terminal subunit S1 protein, which binds to the receptor, and a C-terminal subunit S2 protein, which ensures the fusion of the membranes of the virus and the host cell. The S1 subunit protein consists of two main structural elements: the N-terminal domain (NTD) and the receptor-binding domain (RBD). And it is after RBD binding to the host cell receptor (ACE2) that the sequential cleavage of S-protein at the S1/S2 and S2’ cleavage sites causes an irreversible conformational change in the S2 subunit, which subsequently initiates membrane fusion with the host cell. mAbs directed against SARS-CoV-2 infection block RBD binding to ACE2 and are very effective [40,41,42].

#### 2.1.7. Antibody-Dependent Cellular Cytotoxicity

Antibody-dependent cellular cytotoxicity (ADCC) is a mechanism of adaptive immunity by which nAbs direct cytotoxic effector cells to destroy infected target cells. nAbs bind to viral proteins on the surface of an infected cell. Then, natural killer cells (NK cells) having FcyRIIIa (CD16a, IgG Fc fragment receptor) bind to the antibody Fc fragment [43,44]. This interaction activates the NK cell, triggering a cascade of intracellular ADCC signaling pathways leading to the release of cytotoxic granules containing perforin and granzymes. Perforin forms pores in the membrane of the target cell, allowing granzymes to enter and activate apoptosis (programmed cell death) [45,46].

#### 2.1.8. Antibody-Dependent Cellular Phagocytosis

Antibody-dependent cellular phagocytosis (ADCP) is an immune response mechanism in which phagocytic cells such as macrophages and neutrophils engulf and destroy pathogens or infected cells opsonized by antibodies. In the context of viral infections, ADCP is triggered when antibodies specifically bind to viral antigens present on the surface of virions or infected cells. After binding of antibodies to antigens, the Fc fragment of the antibody interacts with Fc receptors expressed on the surface of phagocytic cells such as FcyRI, FcyRIIA and FcyRIII. This interaction initiates the activation of the phagocyte and the formation of pseudopods that surround the viral particle or infected cell, forming a phagosome. After fusion of the phagosome with the lysosome, a phagolysosome is formed, in which the captured material is degraded and destroyed by lysosomal enzymes and other mechanisms such as the production of reactive oxygen species [47,48,49].

#### 2.1.9. Complement Activation

Complement activation is a complex cascade process of protein activation of the complement system initiated by nAbs, which leads to the elimination of viral particles through two main mechanisms: direct lysis and opsonization. nAbs, by specifically binding to viral antigens on the surface of virions or infected cells, trigger the classical complement activation pathway. At the same time, there are lectin and alternative ways of compliment activation resulting from other interactions [50]. Factors affecting the activation of complement by antibodies include: antibody subclass and isotype, IgD and IgE are believed to be unable to activate complement, but IgM, due to its multimeric nature, is an excellent activator, while IgA activates complement via the lectin pathway. These are also valence and binding affinity, Fc glycosylation, antigen distribution, and clustering [51].

## 3. Mechanisms of nAbs’ Formation

### 3.1. Immune Response to Viral Vectors

In recent years, viral vectors have become essential tools in gene therapy, facilitating the delivery of genetic material into host cells [52]. The most commonly used viral vectors include AAVs, lentiviruses (LVs), and adenoviruses (Ads), each with unique properties that determine their use for various therapeutic approaches [53,54,55].

The choice of vector for gene therapy depends on multiple factors. Each vector has its own advantages and disadvantages that must be considered to achieve specific therapeutic goals [53,54,55]. Although AAVs are the safest, their insert size is limited (maximum 4.7 kb including all regulatory elements and the gene of interest itself), making them less suitable for delivering large genes [56,57]. In such cases, next-generation LV vectors are usually chosen, as they demonstrate adequate safety. However, they are mostly not used for direct gene therapy but rather in cell-mediated approaches, which help reducing immune risks [58,59]. Ad vectors, on the other hand, cause a stronger immune response, since their taxa are highly immunogenic and can cause inflammatory reactions [53,60].

The immune response to viral vectors significantly impacts their efficacy in gene therapy. Upon the first administration of a viral vector, the immune system recognizes it as a foreign agent and initiates an immune response, including the production of nAbs [23].

When a viral vector is introduced into the body, various immune mechanisms are activated [61,62]. The initial response involves activation of the innate immune system, triggering cascades such as interferon production and the activation of macrophages and dendritic cells. These cells play a crucial role in antigen presentation and the subsequent activation of the adaptive immune response [22].

Following viral vector administration or pathogen entry, native or pre-existing B cells are activated. These cells may then differentiate into short-lived plasma cells that secrete antibodies or enter germinal centers, where they interact with antigen-presenting follicular dendritic cells and T follicular helper cells. This interaction facilitates class-switch recombination and affinity maturation through somatic hypermutation [63]. Mature B cells within germinal centers, specialized structures in lymphoid tissues, may remain to undergo further somatic mutations or differentiate into memory B cells and long-lived plasma cells that continuously produce antibodies [64]. Upon re-exposure to the viral vector or pathogen, evolved memory B cells may differentiate into antibody-secreting plasma cells, undergo further evolution and acquire new mutations, and/or be induced to expand into new memory B cell clone [14,65], contributing to the formation of immune memory.

Thus, the process of generating nAbs in response to viral vectors used in gene therapy, is a complex one involving interactions between components of both innate and adaptive immunity. Understanding these mechanisms, including how different viral vector types influence nAb formation, is critical for designing safer and more effective therapeutic strategies. Research conducted in the past five years highlights the importance of thoroughly evaluating the immune response to selected viral vectors for successful gene therapy outcomes [66,67,68].

### 3.2. Immune Response to Therapeutic Genes

Therapeutic genes introduced into patient cells aim to correct genetic defects, restore normal cellular function, or even suppress tumor growth [69,70]. As previously discussed, viral vectors are often employed for gene delivery. However, despite their potential, therapeutic genes themselves can trigger immune responses that significantly impact their efficacy and safety [71].

Introducing therapeutic genes into the body can activate both the innate and adaptive immune systems [72]. The immune response to therapeutic genes is determined by both the genetic material itself and the vectors used for its delivery [73].

Additionally, immune reactions to therapeutic genes can lead to classical adverse effects, such as cytokine storms, which intensify negative outcomes [74]. This highlights the need for accurate monitoring of patients’ immune responses during and after gene therapy. A systematic approach to assessing nAbs levels and their impact on therapeutic outcomes has become an important part of modern clinical trials.

### 3.3. Factors Affecting the Formation of nAbs

Understanding the factors that contribute to nAb formation is crucial for optimizing gene therapy strategies. It has been proven that the dose of administered vector affects immune response intensity [75]. In murine models, higher doses resulted in increased nAb production, whereas lower doses may fail to achieve sufficient transduction of target cells and, therefore, may not provide the desired therapeutic effect. Finding the optimal dose that provides maximum efficacy with a minimum risk of nAbs formation is a key challenge in gene therapy [76].

The route of administration may also affect the likelihood of nAbs formation [77]. Certain delivery methods, such as intravenous administration, may help decrease the immune response, while increasing vector concentration at the target site [78]. Alternative administration routes and vector dose adjustments will be discussed in more detail in the following sections.

Prior exposure to a similar virus or gene increases the likelihood of an immune response. This occurs because the immune system retains ‘memory’ and rapidly reacts to re-exposure to the antigen [79]. Therefore, pre-screening patients for the presence of nAbs before gene therapy is essential [80].

For this purpose, various tests are used to determine the total amount of antibodies. One commonly used method is enzyme-linked immunosorbent assay (ELISA), where serum antibodies against AAV bind to whole capsids or peptides immobilized on a plate, followed by detection using secondary antibodies. ELISA tests are easy to use, stable, reproducible, and have potential for use in commercial projects [57]. Additionally, the enzyme-linked immunospot (ELISPOT) assay is used to assess pre-existing cell-mediated immunity by measuring cytokine levels, such as IFN-γ, secreted by activated antigen-specific B or T cells in the presence or absence of stimulants [81].

Genetic predisposition also plays a role in an individual’s immune response to viral vectors. For example, some people may have a genetic predisposition to a stronger immune response, which increases the likelihood of nAbs formation [82]. Studies suggest that polymorphisms in immune response-related genes can influence the duration of transgene expression and the likelihood of nAbs formation [83].

The formation of nAbs is a complex process affected by multiple factors. Understanding these factors and developing strategies to reduce the likelihood of nAbs formation are key to successful gene therapy.

## 4. nAbs’ Impact on Gene Therapy Efficacy

nAbs act as key elements of the humoral immune response and can significantly affect the efficacy of gene therapy, especially when vector-based systems are used. Moreover, antibodies can induce a more pronounced immune response, which not only reduces therapeutic efficacy but also leads to undesirable side effects, such as limited long-term transgene expression, transduction blockade, and the development of cytotoxic responses [84]. As previously mentioned, it is also important to consider individual factors in nAb activity: pre-existing nAb levels can vary substantially among patients, leading to significant variations in gene therapy outcomes [85].

### 4.1. Reduced Efficacy

A major challenge for researchers is the general decrease in gene therapy efficacy due to various factors [86]. One of the factors affecting patient recruitment for clinical trials and the efficacy of AAV vectors, is the high prevalence of pre-existing neutralizing antibodies against AAV capsids in the human population [87]. These antibodies are formed as a result of prior exposure to wild-type AAV, either through natural infection or cross-reactivity between different serotypes [86]. Antibodies formed following primary vector administration have a lesser impact on transduction, but can completely block the re-administration of AAVs of the same serotype. Serological studies reveal a high prevalence of nAbs: approximately 67% of individuals have antibodies against AAV1, 72% against AAV2 [86], 47% against AAV9, 46% against AAV6, 40% against AAV5, and 38% against AAV8 [88] (Table 1) [89].These findings show the predominance of seropositive patients, which excludes them from gene therapy or prevents repeated administration in cases where initial dosing was insufficient or efficacy decreased over time [90]. In addition, the transplacental transfer of nAbs from the mother should also be considered: for example, AAV9-specific nAbs (primarily IgG-class immunoglobulins) are detectable in newborns up to six weeks of age, but their titers drop to undetectable levels by 4–6 months. In a clinical trial on SMA using AAV9-based Zolgensma, elevated nAb levels against AAV9 were found in 5.6% of infants. Therefore, in real-world clinical practice, it is necessary to assess nAb titers before prescribing AAV-based drugs [91].

Another issue is the broad cross-reactivity of different AAV serotypes, leading to the formation of cross-reactive nAbs in a significant number of individuals (30–60%) [92]. Most patients with high nAb titers against AAV2 also exhibit cross-reactive nAbs to AAV1 [93]. Studies have demonstrated that wild-type AAV infections induce antibodies across all IgG subclasses, with IgG1 being predominant, and IgG levels correlate with nAb titers measured in in vitro neutralization assays [94].

Another important aspect of nAbs’ function is that they act specifically on certain serotypes of viruses (serotype specificity). The efficacy of nAbs depends on the sensitivity and prevalence of specific serotypes. For instance, in patients with high nAb levels, administration of an AAV5 vector expressing human coagulation factor IX (AAV5-hFIX) resulted in sustained transgene expression. The impact on this vector was less pronounced compared to AAV2, indicating that AAV5 was more effective in the presence of nAbs [95]. Another mechanism is AAV opsonization, where phagocytic cells facilitate the rapid clearance of viruses from circulation [96] or alter their distribution within the body [97]. This process occurs through Fcγ receptor interactions, leading to the uptake of vectors by dendritic cells and macrophages, reducing their delivery to target tissues, and potentially triggering inflammatory responses [98].

#### 4.1.1. Neutralization of Viral Vectors Preventing Gene Delivery to Target Cells

Studies indicate that even low nAb titers reduce the efficiency of transgene delivery and expression [99]. It was found that when the nAb titer is 1:1, it limits the process of transmitting information from the AAV vector [100], while at a titer of 1:5, hepatic transduction is completely blocked. Research in this area is ongoing to determine the threshold nAb titer beyond which transgene delivery efficiency is significantly reduced [101]. However, trials initially reported a cell-mediated immune response to AAV2 vectors, limiting transgene expression at doses exceeding 2 × 10^12^ vector particles per kilogram, via portal vein injection [95]. Later findings demonstrated a direct impact on gene transfer itself [102].

In addition to the loss of therapeutic efficacy, the presence of capsid-specific nAbs may also activate the complement system and interfere with transgene delivery. Complement activation leads to interaction with antigens, providing a potent signal through binding to B cell receptors, which lowers the activation threshold and enhances the antibody response, while activating macrophages and cytokines, such as interleukins (IL)-8 and IL-1β [103]. Higher AAV doses correlate with increased complement activation, possibly in an antibody-dependent manner [103]. Studies on mice deficient in CR1/2 or C3 (complement proteins) revealed a reduced ability to mount a humoral response to AAV compared to mice previously infected with wild-type AAV, emphasizing the complement system’s role in generating AAV-specific antibodies [104].

#### 4.1.2. Accelerated Clearance of Viral Vectors from the Body

A special feature of nAbs is their ability to potentially influence vector biodistribution away from target cells by redirecting it to secondary lymphoid organs [105]. In turn, this redirection leads to subsequent uptake of viral vectors by antigen-presenting cells [105], reducing target cell transduction and causing total vector loss. Some studies suggest that vectors may accumulate in the liver for several hours after infusion, rather than being eliminated [105]. However, bnAbs have the greater effect on this displacement, whereas nAbs mainly prevent liver transduction, diverting vectors to the spleen and lymph nodes [92].

It is worth noting that nAbs, through their interaction with the body’s T cells, are able to eliminate cells transduced by AAV, causing inflammation in the target organ and reducing gene transfer efficiency and duration [106].

#### 4.1.3. Induction of Cytotoxic Immune Responses

The adaptive immune system plays a key role in responses to viral gene therapy. Cellular immune response includes the activation of cytotoxic T cells approximately 4–12 weeks after vector administration. Animal model studies indicate that Toll-like receptors (TLRs) contribute to T-cell responses by recognizing AAV capsid antigens [107,108]. This immune response can lead to hepatotoxicity, which has been documented in clinical trials involving several AAV serotypes, including AAV2, AAV8, AAV10, and AAV9 [109].

Innate immune responses to AAV vectors are primarily activated through the TLR9–MyD88 signaling pathway, which induces the production of pro-inflammatory cytokines via nuclear factor κB (NF-κB) activation and the synthesis of type I interferons (IFNs) [110]. The NF-κB alternative pathway also affects transgene expression in AAV-transduced cells [111]. These signals promote the activation of major histocompatibility complex (MHC) genes, as well as the production of pro-inflammatory cytokines and chemokines, including type I and III interferons [85]. Secreted IFNs and cytokines amplify innate immune responses through autocrine and paracrine mechanisms, inducing the expression of interferon-stimulated genes, which inhibit viral replication and spread. Cytokines and chemokines are also critical for initiating effective adaptive immune responses and establishing immune memory [104]. In turn, MHC class II molecules are recognized by CD4+ T lymphocytes, leading to the release of ILs, stimulation of B lymphocytes, and subsequent production of nAbs, eventually reducing the efficacy of vector re-administration [112].

Subsequently, cytokines and chemokines recruit additional pro-inflammatory cells that can interfere with transduction and promote the adaptive immune response. Blocking TLR9 signaling completely suppresses the innate response and reduces adaptive reactions, highlighting its critical role in the immune response to AAV. This cascade of immune events enables B cells to actively produce nAbs [112].

### 4.2. Risks and Side Effects

The immune response to gene therapy is a key factor influencing potential side effects, including fever, fatigue, myalgia, and infection-related complications (Table 2). Initially, it was assumed that innate immunity against AAV vectors was negligible. This misconception stemmed from early studies suggesting that AAV2 vectors induce minimal and temporary innate immune activation, unlike the strong and prolonged adaptive responses triggered by Ad vectors [113]. Therefore, only recently research has begun to uncover the full spectrum of risks associated with AAV administration and its potential adverse effects.

#### Immunopathology and Disease Development

The literature indicates that innate immune responses triggered by complement system activation and high titers of nAbs can lead to severe complications. One such complication is thrombotic microangiopathy (TMA), characterized by thrombocytopenia, hemolytic anemia, and organ damage due to microscopic thrombi formation in capillaries and small arteries, including kidney injury [104,114]. Currently, TMA is considered the most common serious condition in patients receiving high-dose systemic AAV gene therapy. Many patients who developed TMA after AAV infusion required hospitalization and treatment, including red blood cell and platelet transfusions, plasmapheresis, and complement inhibitors [115].

Additionally, cases of acute hepatotoxicity have been reported in patients with SMA receiving Zolgensma [85]. One patient experienced late-onset thrombocytopenia and multiple organ failure, though the underlying mechanism remains unclear [116]. Out of more than 1400 patients treated with Zolgensma, the FDA has reported nine cases of TMA [117]. A recent study also described a case of hemophagocytic lymphohistiocytosis in patient with SMA receiving the therapy [116]. Furthermore, three cases of severe hepatobiliary disease were recently documented in patients with X-linked myotubular myopathy (XLMTM), who received a high dose (3 × 10^14^ genome copies/kg) of AAV8 expressing the therapeutic *MTM1* gene [118]. Other adverse events of varying severity have been reported in patients receiving high systemic doses of AAV for the treatment of SMA type I, XLMTM, and Duchenne muscular dystrophy [3].

**Table 2 ijms-26-05224-t002:** Summary Table: Viral vectors used in clinical trials, as well as their potential interaction with nAd and impact on therapeutic efficacy and side effects.

Viral Vector Type	Examples of Applications	Potential Role of Neutralizing Antibodies	Side Effects and Limitations	Sources
**Adeno-Associated Viruses**	Gene therapy for inherited diseases (e.g., spinal muscular atrophy, hemophilia), oncology.	nAbs to AAV can reduce therapy effectiveness by preventing vector entry into target cells. Patients with pre-existing antibodies to AAV may require vector serotype switching.	Low immunogenicity, but inflammatory reactions in the liver are possible. Limited genome capacity (~4.9 kb).	[89,119,120,121,122]
**Lentiviruses**	Treatment of hemoglobinopathies (e.g., β-thalassemia), CAR-T therapy.	Lentiviruses are less susceptible to neutralization by antibodies but may elicit an immune response to viral proteins.	Risk of insertional mutagenesis due to integration into the host genome. Oncogene activation is possible.	[123,124,125,126,127,128]
**Retroviruses**	Ex vivo therapy (e.g., treatment of SCID), oncology.	Neutralizing antibodies can limit repeated vector administration.	Infect only dividing cells. High risk of insertional mutagenesis.	[129,130,131,132,133]
**Adenoviruses**	Vaccines (e.g., against COVID-19, HPV), oncology.	High levels of pre-existing antibodies to adenoviruses in the population can reduce effectiveness.	Strong immune response, risk of cytokine storm. Limited use upon re-administration.	[134,135,136,137,138]
**Herpes Simplex Viruses**	Neurodegenerative diseases, oncology.	Antibodies can reduce delivery efficiency.	Ability to infect only non-dividing cells. Neurotoxicity, inflammatory reactions in the CNS are possible.	[139,140,141,142]

### 4.3. Complications Associated with Repeated Injections

As previously mentioned, naturally occurring nAbs interfere with the systemic delivery of AAV vectors [143]. This poses significant challenges for the long-term success of gene therapy, mainly in patients receiving low doses of AAV, such as children, as tissue growth and proliferation may dilute the vector genome, potentially worsening symptoms over time. This also applies to patients with degenerative disorders who may require repeated AAV treatments to prevent tissue loss and maintain therapeutic transgene expression levels [144]. Administering gene therapy to patients with pre-existing nAbs can also trigger systemic inflammatory responses due to immune complex formation, increased vector uptake by antigen-presenting cells, and complement activation.

#### 4.3.1. Reduced Therapeutic Efficacy Due to Elevated nAb Levels

Numerous studies have demonstrated a correlation between high pre-existing serum antibody levels and reduced AAV vector transduction efficiency upon re-administration [114]. However, in immune-privileged tissues, such as the eye, pre-existing humoral immunity has shown minimal impact on vector delivery, allowing for repeated treatment in the contralateral eye [145].

There are conflicting data on the relationship between pre-existing anti-AAV nAb titers and vector transduction efficiency. Despite the sharp increase in antibody titers following vector administration, gene therapy outcomes can still meet expected therapeutic levels. For example, in the hemophilia B clinical trial mentioned earlier, one of the serotypes used for the vector was AAV2. Following AAV-hFIX administration, anti-AAV titers increased more than 10,000-fold. However, therapeutic levels of FIX were still achieved, with a decline observed approximately eight weeks post-infusion, consistent with the development of capsid-specific T cells [95,112].

Additionally, there have been multiple studies conducted to alter the concentration of nAbs produced. For instance, in rodent experiments, different AAV delivery vectors were used for the second administration, reducing cross-reactive antibodies and lowering nAb levels [114]. However, similar studies in non-human primates showed that nAb reactivity against AAV5, AAV8, or AAV9 remained largely unchanged following the administration of both wild-type and mutant AAV2 capsids, such as 4YF or 7m8 [114].

#### 4.3.2. Increased Risk of Adverse Effects

One proposed solution to the problem of repeated dosing is a single high-dose administration of the gene therapy. However, growing evidence suggests that high vector doses and excessive empty capsid content cause dose-dependent immune toxicity. This highlights the need to reduce the interaction between the AAV vector and the immune system, potentially requiring a reassessment of the single high-dose treatment approach [145,146]. In other words, the resulting toxicity may limit the duration of transgene expression, for example, in hepatocytes.

The liver is the primary target organ for systemically delivered AAV, and immune-mediated liver toxicity can occur at any time following high-dose vector administration [145]. Manufacturing methods also contribute to certain risks associated with impurities and variations in their concentrations. Some of these impurities may act as immunogenic elements, including excess non-infectious capsids and contaminants.

Vector design plays as important a role, as its dose, in determining immunogenicity. Emerging data from recent clinical trials and preclinical models suggest that CpG-rich sequences (cytosine-phosphate-guanine motifs) in the vector genome significantly enhance its immunogenicity. In recent years, studies have reported novel AAV-associated inflammatory responses in non-human primates and neonatal piglets [104]. These responses include neuroinflammation in dorsal root ganglia following intravenous high-dose administration, as well as acute thrombocytopenia and liver and kidney toxicity following similar administration [147].

## 5. Strategies to Overcome Challenges Associated with nAbs

### 5.1. Surface Modification of Viral Vectors to Reduce Immunogenicity

One example of modifying viral vectors to reduce immunogenicity is the use of insect baculovirus (BV) vectors, which are widely employed in many vaccines. For instance, unlike bacterial expression systems, BV systems support post-translational modifications, enabling the production of mammalian proteins in their native form. In general, the baculovirus display system is based on the natural ability of these insect viruses to express and display proteins on the surface of baculovirus particles. This system makes it possible to embed the gene encoding the protein of interest into the genome of the virus, which ensures its synthesis within infected host cells, frequently insect cells such as Spodoptera frugiperda (Sf9). Peptides fuse with viral envelope proteins (for example, gp64), which allows them to be displayed on the surface of the virus, while the system remains safe for humans [148]. Compared to AAV vectors, which have a limited gene insertion capacity (~4.7 kb), BVs can take large (>100 kb) foreign DNA fragments. Since BV does not require helper viruses for replication, recombinant virus generation is relatively quick and straightforward [149]. Findings from one study could lead to advancements in the baculovirus display system, and provide an experimental basis for the development of insect BV vectors for many vaccines. A recombinant BV vector has been developed that exhibits enhanced display efficiency, higher viral titers than previous systems, and improved BV transduction efficiency in mammalian cells [150].

The high stability of the AAV capsid under extreme conditions has facilitated the development of methods to modify its surface through cross-linking with various chemical molecules. Initially, peptides with an RGD structure; a common recognition and protein–protein interaction motif in cellular proteins, were proposed, along with polyethylene glycol, to improve AAV tropism and reduce the impact of nAbs on transduction. Although chemical modification of AAV vectors can theoretically significantly alter its surface, only partial reduction in nAbs’ effect has been reported so far. One major limitation of chemical capsid modification, and protein modification in general, is that achieving substantial surface changes requires extreme conditions that may compromise the stability of the virus’s three-dimensional structure [6].

Moreover, systemic administration of AAV often inactivates NAbs and high doses cause hepatotoxicity. To address these issues, ternary AAV complexes based on tannic acid and boronate-containing polymers were developed. In these complexes, AAV9 (average diameter 25 nm) is encapsulated in a protective shell, forming particles of 60 nm in size. Tannic acid opens the AAV capsid and forms boronate esters with the polymers, creating a barrier against NAbs. Intravenously administered complexes avoid neutralization and reduce hepatotoxicity by reducing accumulation in the liver [151].

Recent studies have applied a method of randomly generating chimeric capsids (capsid shuffling) based on the primary sequences of known serotypes. When combined with error-prone PCR, this approach enables the creation of highly diverse capsid libraries, facilitating the exploration of a wider range of capsids with inherently different tropism profiles [84]. A single chimeric capsid formed from five different parental AAV capsids was found to exhibit high transduction efficiency in primary human hepatocytes both in vitro and in vivo, providing species-specific transduction in liver cells in vitro. This vector shows promise for transduction and genetic modification of xenograft cells in mouse models of human diseases [152].

### 5.2. Immunomodulation

#### Immunosuppressive Therapy and Induction of Immune Tolerance

Despite the active development of gene therapy using AAV, the body’s immune responses remain a serious obstacle to its widespread use. Current strategies to combat this include not only modification of the AAV capsid and optimization of the viral genome, but also the use of immunosuppressive therapy [153].

Corticosteroids such as prednisolone and methylprednisolone are widely used to suppress the immune response to AAV. They reduce the level of pro-inflammatory cytokines and minimize liver damage. Initially, they were prescribed reactively—in response to an increase in liver enzymes, which was considered a sign of a T-cell response to the viral capsid. However, a preventive approach is now more commonly used: in 74% of studies, immunosuppression is started 1–3 days before vector administration [153]. In addition to corticosteroids, calcineurin inhibitors (e.g., tacrolimus) are used, which suppress T-cell activation and interleukin-2 production, which indirectly affects the B-cell response and the level of neutralizing antibodies. In some cases, combination regimens are used, including rituximab, sirolimus, and corticosteroids, which have shown effectiveness, for example, in the treatment of GM2 gangliosidosis [102,154].

However, long-term immunosuppression increases the risk of infections and viral reactivation, which is especially dangerous for patients with severe diseases. In addition, in some cases, the phenomenon of immune tolerance is observed, when the body does not respond to the administered AAV vector. Regulatory T cells (Treg) play an important role in this process, suppressing the activity of effector immune cells [155,156].

An example of successful application of AAV therapy is the treatment of hemophilia B using the scAAV2/8-LP1-hFIX vector. A single administration ensured long-term expression of coagulation factor IX without significant toxicity. A short course of glucocorticoids allowed to control the immune response to the capsid and normalize the level of liver enzymes without disrupting the expression of the therapeutic gene. This approach demonstrates the potential of gene therapy for radically improving the condition of patients with severe forms of the disease [102].

However, safety issues and dosage optimization require further study. The development of new strategies, including immune tolerance induction and vector engineering, may help overcome existing limitations [157].

### 5.3. Alternative Strategies for Preventing nAb Functioning

#### 5.3.1. Vexosomes and Extracellular Vesicle-Associated AAVs

One potential strategy to physically prevent nAb-mediated neutralization involves associating AAV with microvesicles/exosomes (vexosomes) [158]. Vexosomes are a type of extracellular vesicles derived from cells infected with a virus. Their mechanism is based on encapsulating AAV particles within or on the surface of exosomes. This encapsulation can shield AAV from immune detection and facilitate cellular uptake through exosome-mediated pathways, potentially enhancing delivery efficiency and tissue targeting, which allows for more efficient gene delivery even in the presence of pre-existing immunity [159].

To obtain these structures, AAV vector production was modified so that the resulting vectors are coupled to the surface and interior of microvesicles, forming vexosomes. This was achieved by pelleting AAV-associated microvesicles by differential centrifugation of supernatant from a production bioreactor. Although the mechanism by which AAV-associated microvesicles are transported from the cell surface to the nucleus remains unknown, they successfully transduced cells even in the presence of sustained nAb concentrations significantly exceeding levels typically observed in patients [145]. Moreover, studies show that vexosomes significantly improve gene transfer efficiency compared to conventional AAV vectors, both in vitro and in vivo, across various cell types and tissues, including lung cancer and liver cells [160]. In the article by Meliani A. et al., a 40-fold increase in the transduction of exo-AAV5 was recorded compared to standard AAV5, and complete restoration of coagulation activity was observed in mice with hemophilia B. This was achieved using low doses of exo-AAV8 vectors, which were compared to AAV8. Additionally, these exo-vectors were able to stimulate higher levels of transgene expression [161]. The significant advantage of this strategy lies in the fact that vexosomes could allow gene therapy in patients who would otherwise be excluded due to pre-existing immunity. In the current research, these delivery method is considered as a more optimal option, because it resulted in greater therapeutic benefits, for instance it had enhanced tumor regression in hepatocellular carcinoma, improved cardiac function post-myocardial infarction, and restoration of sensory function in hearing loss models [162].

#### 5.3.2. Degradation of nAbs by Pre-Injection of Enzymes That Break Down Ig

Pre-injection of IgG-cleaving enzymes to eliminate circulating nAbs before the introduction of the AAV vector is a strategy aimed at increasing the effectiveness of gene therapy [163]. The choice of IgG as a target for enzymatic degradation is determined by several key factors. IgG is the predominant class of antibodies in human and other mammalian blood serum, accounting for a significant portion of the total number of immunoglobulins. Due to their high concentration, IgG is most likely involved in neutralizing AAV vectors. In addition, IgG is characterized by a long half-life (about 21 days in humans), which provides prolonged protection against infections. However, the same factor causes the prolonged presence of IgG nAbs in the circulation, preventing effective gene therapy [164]. Although IgM and IgA may also have some neutralizing activity, their contribution to the neutralization of AAV is usually less significant compared to IgG [88].

The IDEs drug is a possible escape from neutralizing antibodies. It is a bacterial pathogen Streptococcus pyogenes protease that cleaves human IgG antibodies in the hinge region, resulting in the formation of Fc and Fab fragments, the main function is to protect the bacterium from phagocytic destruction [165]. After just a few minutes of administration to humans, IgG worsened the effector functions, reducing binding to Fcyy receptors and Fc-mediated cytotoxicity, the effect persisted for several weeks [166].

Another IdeZ drug, a homolog of IDEs, was identified and characterized in Streptococcus equi subsp. zooepidemicus has shown the same effect on IgG in human serum, non-human primates. The use of other methods that promote the elimination of IgG (soluble bacterial proteins binding antibodies; immunosuppressive drugs) can be used in parallel, complementing the approach to IgG degradation mediated by IdeZ and IDEs. One possible disadvantage of this approach is that humans can produce antibodies against drugs [143].

Modified enzymes that simultaneously cleave IgM and IgG have been reported to reduce antibody neutralization and complement activation during AAV gene transfer. An alloy enzyme (IceMG) with dual proteolytic activity against these immunoglobulins caused by AAV9 capsids has been created. The authors suggested that pre-administration of IceMG 1–2 days before the administration of AAV may provide the possibility of effective suppression of nAd-virus complement activation in patients [167].

### 5.4. Modification of Delivery Methods

Furthermore, vector administration routes influence nAb impact: after intravenous administration, the effect of nAbs on transgene expression is more pronounced, whereas such effects are significantly reduced with intramuscular delivery or subretinal administration (where limited blood contact minimizes effects on AAV-mediated transgene transfer) [87]. Studies using intrathecal administration have also shown a significant reduction in the attenuating effect of nAbs, for example, when the vector is administered intrathecally for the treatment of spinal muscular atrophy, it is delivered directly into the cerebrospinal fluid. The level of nAb in the cerebrospinal fluid is significantly lower than in peripheral blood, increasing the chances of successful neuronal transduction. But there is still a possibility of inducing an immune response to the therapeutic product due to local immunogenicity [168].

In the case of intramuscular injections, where the vector can drain into the lymphatic system and provoke an immune response, it is important to optimize the dosage and volume of the injection in order to minimize leakage of the vector into the systemic circulation. Although many studies emphasize the advantages of intramuscular administration, as it ensures successful gene expression even in the presence of high levels of nAb, systemic administration (e.g., intravenous) is blocked by nAbs [169].

In addition, some tissues, such as the liver, remain highly accessible to antibodies even with local administration due to their abundant vascularization, which requires additional strategies to overcome nAt, such as temporary immunosuppression or the use of engineered capsids. So, when AAV is delivered systemically for liver targeting, even low NAb titers can completely block gene transfer, and alternative delivery routes do not consistently overcome this barrier [87].

The eye is a highly compartmentalized organ with an anatomically isolated local environment, shielded by the ocular vasculature and the absence of lymphatics. Therefore, it has long been thought that the eye is at lower risk of innate and adaptive immune activation following genetic vector administration. Furthermore, a phenomenon known as associated with anterior chamber (the area corresponding to the transition of the cornea into the sclera and the iris into the ciliary body) immune deviation has been described in the eye, which involves the induction of Tregs, anti-inflammatory (M2) macrophages, and the formation of an anti-inflammatory cytokine environment that maintains immunological tolerance [170]. This pro-tolerogenic environment serves as a protective adaptation to prevent inflammatory responses that could impair vision. A similar immune deviation phenomenon has been reported following AAV vector administration into the subretinal space.

The high degree of immune privilege in the eye is reflected in the fact that circulating AAV capsid antibodies are generally not detected in ocular structures. Additionally, the low vector doses required to achieve therapeutic effects have facilitated successful gene transfer following intraocular AAV vector administration. Based on these promising findings and more than two decades of research in ocular gene therapy, this field has expanded significantly in recent years, with numerous gene transfer trials planned, mostly using AAV vector [110]. However, it is important to note that intravitreal administration is more susceptible to nAbs, leading to reduced gene transfer and increased immune response, as the vector is more exposed to the immune system in this route. Moreover, pre-existing nAbs can increase the risk of post-operative inflammation after intravitreal injection [114].

To date, AAV vector administration has predominantly been performed via intravitreal or subretinal injection. The immunogenic response largely depends on the route of vector delivery to the eye. However, depending on the total administered vector dose, inflammatory reactions may still occur, regardless of the delivery method. Gene therapy for ocular diseases continues to evolve and new technologies and advances may shape the future direction of the field [110,171].

### 5.5. Selection of Optimal Doses

Clinical trial experience suggests that AAV vector immunogenicity is, to some extent, dose-dependent, with low vector doses more likely to induce mild and manageable inflammation that does not result in complete transgene expression loss. This is supported by in vitro studies showing dose-dependent levels of capsid antigen presentation on MHC class I molecules of transduced cells. Also, when the dose of AAV is reduced, the number of viral particles available for binding to nAb decreases. Therefore, in cases where antibody titers are not too high, some of the vectors may be able to avoid neutralization and reach target cells, providing some level of transduction. Additionally, a lower antigenic load may theoretically reduce the activation of B cells and the production of new nAbs. However, if nAb levels are initially high, even a significant reduction in dose may not be effective, as the antibodies will continue to bind and neutralize the majority of viral particles [172].

On the other hand, increasing the dose has sometimes been considered as a way to “overcome” nAb due to the large number of vectors that can saturate antibodies and provide a sufficient number of free particles for transduction. However, this approach carries significant risks, including an increased immune response, potential toxicity, and the possibility of a sudden increase in nAb titers after administration, which could completely block subsequent treatment attempts [173].

When selecting the optimal dose for treatment, the individual’s immune status remains a crucial factor. At lower or moderate antibody levels, a reduction in the dose may be partially effective. However, at higher levels, additional strategies may be required, as discussed previously.

Additionally, the treatment of some monogenic diseases necessitates very high doses, which can cause serious adverse effects. These effects are observed with both systemic and local injections of central nervous system serological AAV vectors. Recent research suggests that the safe dose for some conditions is lower than the therapeutic dose. This emphasizes the need to continue investigating immunosuppressive approaches that can inhibit B-cell responses induced by AAV treatment, thereby opening the door to repeated injections of safer doses [174].

Other factors influencing vector immunogenicity are less well known and may include pre-existing tissue inflammation, the use of single- or double-stranded vector genomes, and the CpG content of the vector genome [110]. However, more research is needed for better understanding of the immune response to viral vectors. This knowledge could help to specifically modify the immune response, increasing vector efficacy and allowing for repeated dosing [175].

## 6. Perspectives and Future Research

### 6.1. Development of New, Safer Viral Vectors

An Ad-free method for generating AAVs relies on the transient transfection of mammalian cells with three plasmids. Two of these plasmids provide the in trans genes *rep* and *cap*, as well as auxiliary genes, typically from Ad. The third plasmid contains a transgene expression cassette flanked by two inverted terminal repeats. AAV vectors can also be produced in mammalian cells by infecting them with Ad or herpes simplex virus. Additionally, AAV can be produced in BV-infected insect cells, which carry all necessary components for vector production. Regardless of the chosen production method, and unlike wild viruses, AAV vectors are formed as a mixture of complete capsids with genomic material as well as empty capsids [6].

### 6.2. Development of New Immunomodulation Methods

Immunomodulation regimens aim to prevent the T-cell immune response to AAV. These regimens are typically difficult to capture in clinical trials, though 38% of them indicate a potential for the use of immunosuppressants. Additionally, 45% of trials exclude participants who are on (chronic) immunosuppressive therapy, immunotherapy, or have immunodeficient conditions. Although the exact use of immunosuppression and its consequences on clinical outcomes remain largely unknown due to inadequate reporting and unpublished data, 38% is a high enough figure to suggest the potential value of this strategy in preventing immune responses against AAV [176].

The use of immunosuppressants in clinical trials is poorly documented, and the lack of consistency in published data makes the effective methods to prevent anti-capsid cellular immunogenicity difficult to find. Although immunosuppression with oral steroids has shown success in limiting transgene expression loss in some AAV hepatocellular gene therapies, other studies have demonstrated that steroids alone are insufficient to protect against transgene loss. This creates a knowledge gap that requires additional clinical data to form conclusions and design future clinical trials [177].

### 6.3. Investigating Individual Differences in Immune Response to Gene Therapy

A study of 125 healthy adult blood donors from the United States, with a median age of 43 years, found pre-existing nAbs against SaCas9 (Staphylococcus aureus Cas9) and SpCas9 (Streptococcus pyogenes Cas9) in 78% and 58% of samples, respectively. In addition to antibodies, the study also assessed pre-existing cellular immune responses. The authors observed an increase in the number of cells releasing IFN-γ in peripheral blood mononuclear cells from 18 donors stimulated with SaCas9 and SpCas9 [178]. Among the donors, 78% had SaCas9-specific T-cells, and 67% had SpCas9-specific T-cells. Cytokine-positive cells were detected using intracellular staining targeting IFN-γ, tumor necrosis factor (TNF)-α, and IL-2. Activated T-cells were identified by FACS (Fluorescence-activated cell sorting) sorting using activation markers CD137 and CD154. Another study conducted on 48 healthy donors also confirmed a high prevalence of SpCas9-specific T-cells, showing that SpCas9 stimulation could activate CD137+ and CD154+ T-cells. A study comparing existing anti-SaCas9 and anti-SpCas9 antibodies in serum and vitreous fluid from 13 patients who underwent vitrectomy surgery found that all three serum samples tested positive for SaCas9 and SpCas9, whereas only two vitreous fluid samples were positive for SaCas9, and two others for SpCas9. This study also showed that nAb levels were higher in serum compared to vitreous fluid [172].

In a study comparing the transduction efficiency and safety of various viral vectors, green fluorescent protein (GFP) packaged in AAV, LV, Ad, and BV was intravitreally injected into C57BL/6OlaHsd mice. Using immunostaining of GFP in retinal sections, the study showed that LV could provide prolonged GFP transgene expression in RPE, which was more pronounced than Ad expression but less intense than AAV. Mice injected with BV showed no GFP-positive cells after 7 days. However, evaluation of inflammatory reactions and toxicity revealed that LV promoted macrophage recruitment and the production of anti-transgene antibodies, but not as much as Ad. When comparing F4/80-positive cells (a primary macrophage marker) in retinal sections of mice injected with Ad or BV, the highest levels of F4/80-positive cells were observed, followed by LV, while the lowest levels were seen in the AAV group 3 and 7 days post-injection. Additionally, GFP antibody levels measured by ELISA in serum samples showed that Ad induced the highest levels of antibodies, followed by LV, BV, and AAV [179].

## 7. Conclusions

Gene therapy represents a promising approach for treating a wide range of diseases, including genetic disorders and cancer. However, the formation of nAbs against viral vectors used for delivering therapeutic genes presents a major challenge to its broad application. nAbs can neutralize viral vectors, preventing their entry into target cells and reducing therapy efficacy.

Understanding the mechanisms behind nAb formation and the factors that influence their production is the key to developing strategies to overcome this challenge. Research indicates that the dose of the administered vector, the method of delivery, prior immunization, and the patient’s genetic predisposition may all impact the immune response and the possibility of nAb formation.

It is important to note that nAbs can not only reduce therapy effectiveness, but also cause side effects, including immunopathology, cytokine storms, and complications with repeated injections.

Despite existing challenges, gene therapy remains a promising approach for treating a broad spectrum of diseases. Further research aimed at understanding the mechanisms of nAb formation and developing strategies to neutralize them will undoubtedly lead to the development of safer and more effective gene therapy methods in the future.

## Figures and Tables

**Figure 1 ijms-26-05224-f001:**
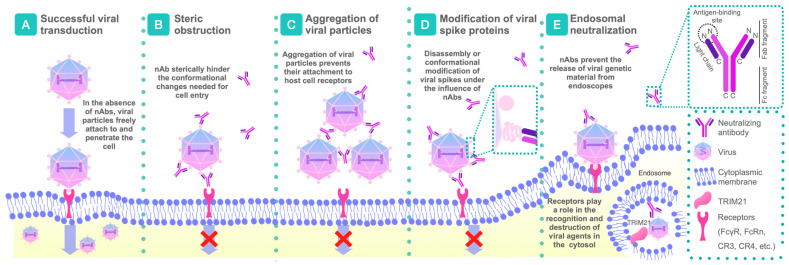
nAbs’ mechanisms of action. (**A**) Successful viral transduction and attachment in the absence of nAbs. (**B**–**E**) Four primary mechanisms of action of nAbs. (**B**) Steric obstruction after viral attachment. Fusion of the viral and host cell membranes is blocked when a bulky antibody molecule interacts with the relevant viral site. The steric blockade can be enhanced by binding of the complex to the FcyR receptor on immune cells. (**C**) Aggregation of virions by nAbs prevents their attachment to host cell receptors. It can be enhanced by cross-binding of antibodies to the FcR receptor on the cell membrane. (**D**) Disruption or conformational modifications of viral spikes. Neutralization occurs when an antibody occupies a substantial number of accessible epitopes on the virion surface, leading to the inhibition of viral adhesion and impairing its entry. (**E**) Endosomal neutralization is carried out through the interaction of nAbs with the FcyRIIB receptor. nAbs can neutralize viruses within endosomes by preventing the release of viral genetic material, which obstructs its integration and replication within host cells. Intracellular antibody activity also includes interactions with receptors such as TRIM21.

**Table 1 ijms-26-05224-t001:** Summary Table: Global Seroprevalence of NAbs to Common AAV Serotypes.

Serotype	Global Seroprevalence (%)	Notable Trends/Populations	Geographic Variability
**AAV2**	58–97% (72% average)	Highest overall, increases with age	High across all regions
**AAV1**	67%	2nd highest, co-prevalent with AAV6	Variable by country
**AAV5**	7–35%	Lowest, increases with age	5.9% (UK) to 51.8% (S. Africa)
**AAV6**	20–49%	Intermediate, co-prevalent with AAV1	Variable
**AAV8**	7–46%	Lower in children	Variable
**AAV9**	1–56%	Lower in children, higher in adults	Variable

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
