# Peer review of "Neutralizing Antibodies: Role in Immune Response and Viral Vector Based Gene Therapy"

_ijms, 2025, doi:10.3390/ijms26115224_

Round 1

Reviewer 1 Report

Comments and Suggestions for Authors

The authors have summarized the impact of neutralizing antibodies on the efficacy of AAV based gene therapy. The review is nicely written and well organized. Few suggestion to further improve the manuscript are provided below. 

  1. Please consider changing the title of the review to " Neutralizing antibodies: Role in immune response and viral vector based gene therapy"
  2. In section 2.1, authors have listed various neutralization mechanisms of nAbs. It will be beneficial to add more examples for the discussed mechanisms to get more insight into their importance. 
  3.  A graphic representation of various Immune pathways activated upon AAV/ viral vector administration will be helpful for section 3.1. 
  4. I would recommend adding a table of various viral vectors previously or currently being evaluated in clinical trials and potential role of nAbs on therapeutic efficacy and side effects, if reported. 
  5. The review is quite long with some general statements being repetitive. I would recommend to focus on scientific facts and making the article concise. 
  6. In section 5.2.1, role of immune tolerance as immunosuppressive therapy for viral vectors is unclear. I would recommend adding more clinically relevant information in addition to just describing the phenomenon. 
  7. Section 5.3 titles seem incorrect. Both 5.3.1 and 5.3.2 are not about neutralizing nAbs but alternate strategies to prevent nAb functionality by interfering with its interaction to the target antigens. 

Author Response

We thank the reviewer for the valuable comments. We have improved the manuscript according to the suggestions.

Comments 1: Please consider changing the title of the review to "Neutralizing antibodies: Role in immune response and viral vector based gene therapy"

Response 1: We agree that the proposed title, “Neutralizing antibodies: Role in immune response and viral vector based gene therapy” more accurately reflects the focus of our review, and have decided to make this correction.

Comments 2: In section 2.1, authors have listed various neutralization mechanisms of nAbs. It will be beneficial to add more examples for the discussed mechanisms to get more insight into their importance. 

Response 2: We fully agree that adding additional examples for various neutralization mechanisms will help to better reveal their importance. In this regard, we have significantly expanded section 2.1 to include new examples. We hope that this will improve the understanding of the presented material.

Comments 3: A graphic representation of various Immune pathways activated upon AAV/ viral vector administration will be helpful for section 3.1.

Response 3: We appreciate your suggestion to include a graphic representation of various immune pathways activated upon AAV/viral vector administration in Section 3.1. However, in the current version of the manuscript, we do not explore the detailed mechanisms of these immune pathways. For this reason, we believe that adding such a figure may go beyond the scope of the discussion and could distract from the main narrative of the section. We hope for your understanding on this point.

Comments 4: I would recommend adding a table of various viral vectors previously or currently being evaluated in clinical trials and potential role of nAbs on therapeutic efficacy and side effects, if reported. 

Response 4: We have added the “Summary Table: Viral vectors used in clinical trials, as well as their potential interaction with nAd and impact on therapeutic efficacy and side effects” table according to your recommendation.

Comments 5: The review is quite long with some general statements being repetitive. I would recommend to focus on scientific facts and making the article concise. 

Response 5: We appreciate your opinion regarding the length of the review. We have structured the article in such a way as to present the topic in a logical and understandable way, starting with general principles and gradually moving on to more detailed aspects. We believe that each of the provisions presented is important for a complete understanding of the topic, and therefore the current length is reasonable.

Comments 6: In section 5.2.1, role of immune tolerance as immunosuppressive therapy for viral vectors is unclear. I would recommend adding more clinically n.

Response 6: We have improved section 5.2.1 and clinical cases have been added

Comments 7: Section 5.3 titles seem incorrect. Both 5.3.1 and 5.3.2 are not about neutralizing nAbs but alternate strategies to prevent nAb functionality by interfering with its interaction to the target antigens.Response 7: We agree that the current title of the chapter doesn't fully capture the essence of the information being discussed. Therefore, we have changed the title of this section to reflect this, combining chapters 5.3.1 and 5.3.2. It also has allowed us to explore more alternative strategies for overcoming the problem of nAb.

Reviewer 2 Report

Comments and Suggestions for Authors

Albert A. Rizvanov and his team presented a manuscript entitled “Neutralizing Antibodies: Their Role in Immune Response and Impact on Gene Therapy Efficacy.” This is a highly relevant and engaging topic. The authors provide a comprehensive summary of the mechanisms behind the formation of neutralizing antibodies (nAbs) against adeno-associated virus (AAV) vectors. Additionally, they discuss various strategies to overcome or mitigate the effects of nAbs to improve the efficacy of gene therapy.

Please address the following concerns and comments.

  1. Figure 1: Indicate which cell surface receptors are involved in the recognition and capture of the nAb–virus complex—for example, FcγRIIb.
  2. Section 5.3.2, titled "Antibodies Capable of Neutralizing nAbs," does not actually discuss any antibodies that are capable of neutralizing neutralizing antibodies.
  3. Define Vexosomes. Vexosomes are a type of extracellular vesicle derived from virus-infected cells. We recommend revising and expanding this section to provide a clearer interpretation, supported by recent findings or hypotheses.
  4. Degradation of nAbs by Pre-injection of Ig-Cleaving Enzymes:
    Several IgG-degrading enzymes have been developed to eliminate pre-existing circulating neutralizing antibodies (nAbs) prior to AAV vector administration, thereby enhancing the efficacy of AAV-based gene therapy. We recommend that the authors include a dedicated section discussing this approach, as it represents a promising strategy for overcoming one of the key barriers to successful gene delivery.
  5. A recent study addressed the issue of neutralizing antibodies (nAbs) by coating AAV9 vectors with polymers. We recommend that the authors include a discussion of this approach, as it provides a promising strategy for overcoming nAb-mediated challenges in gene therapy. For instance, Tannic acid-coated AAV vectors form a ternary micelle when combined with phenylboronic acid-conjugated polymers. This ternary complex is successfully packaged into a core compartment, surrounded by polymer chains that create a protective shell, helping to evade inactivation by neutralizing antibodies (nAbs). Upon intravenous injection, these ternary complexes effectively evade nAb-mediated neutralization and reduce hepatotoxicity by minimizing liver accumulation (ACS Nano 2025;19(8):7690-7706). Please add this information to Section 1. Surface modification of viral vectors to reduce immunogenicity
  6. How do Sections 5.3.1 ("Antibodies Blocking nAbs") and 5.3.2 ("Antibodies Capable of Neutralizing nAbs") differ? It is unclear with the current style description. Please improve the discussion.
  7. Please include a discussion of existing studies on the estimated global seroprevalence of neutralizing antibodies (nAbs) against common AAV serotypes.
  8. Does decreasing the dose of AAV and altering the route of administration help overcome the issue of neutralizing antibodies (nAbs)? If so, please provide a detailed discussion.

Author Response

We thank the reviewer for the valuable comments. We have improved the manuscript according to the suggestions.

Comments 1: Figure 1: Indicate which cell surface receptors are involved in the recognition and capture of the nAb–virus complex—for example, FcγRIIb. 

Response 1: Changes have been made to Figure 1, displaying receptors on the cell surface involved in the recognition and capture of the nAb-virus complex. In particular, the FcγRIIb receptor was added.

Comments 2: Section 5.3.2, titled "Antibodies Capable of Neutralizing nAbs," does not actually discuss any antibodies that are capable of neutralizing neutralizing antibodies.

Response 2: We agree with this remark, therefore sections 5.3.1 and 5.3.2 were combined under the general name “Alternative strategies for preventing nAbs functioning”, and the content was adjusted according to the comments of the reviewers.

Comments 3: Define Vexosomes. Vexosomes are a type of extracellular vesicle derived from virus-infected cells. We recommend revising and expanding this section to provide a clearer interpretation, supported by recent findings or hypotheses.

Response 3: We have provided a definition of vexosomes and have reviewed in more detail the current research findings on this topic in Section 5.3.1.

Comments 4: Degradation of nAbs by Pre-injection of Ig-Cleaving Enzymes: Several IgG-degrading enzymes have been developed to eliminate pre-existing circulating neutralizing antibodies (nAbs) prior to AAV vector administration, thereby enhancing the efficacy of AAV-based gene therapy. We recommend that the authors include a dedicated section discussing this approach, as it represents a promising strategy for overcoming one of the key barriers to successful gene delivery.

Response 4: We have included a separate section dedicated to the degradation of nAbs by pre-administration of IgG-cleaving enzymes, according to your suggestion.

Comments 5: A recent study addressed the issue of neutralizing antibodies (nAbs) by coating AAV9 vectors with polymers. We recommend that the authors include a discussion of this approach, as it provides a promising strategy for overcoming nAb-mediated challenges in gene therapy. For instance, Tannic acid-coated AAV vectors form a ternary micelle when combined with phenylboronic acid-conjugated polymers. This ternary complex is successfully packaged into a core compartment, surrounded by polymer chains that create a protective shell, helping to evade inactivation by neutralizing antibodies (nAbs). Upon intravenous injection, these ternary complexes effectively evade nAb-mediated neutralization and reduce hepatotoxicity by minimizing liver accumulation (ACS Nano 2025;19(8):7690- 7706). Please add this information to Section 5.1. Surface modification of viral vectors to reduce immunogenicity

Response 5: We added this information to Section 5.1. Surface modification of viral vectors to reduce immunogenicity

Comments 6: How do Sections 5.3.1 ("Antibodies Blocking nAbs") and 5.3.2 ("Antibodies Capable of Neutralizing nAbs") differ? It is unclear with the current style description. Please improve the discussion. 

Response 6: We agree that the previous section titles may be misleading. Therefore, we combined these sections in a new table of contents and focused on discussing alternative ways to overcome nAb, which are more consistent with the information originally discussed in these chapters.

Comments 7: Please include a discussion of existing studies on the estimated global seroprevalence of neutralizing antibodies (nAbs) against common AAV serotypes.

Response 7: We focused on this topic in Section 4.1. However, for the convenience of readers, we’ve added a table reflecting current values for the global seroprevalence of AAV serotypes. This table is based on the most recent data available from the 2024 study. 

Comments 8: Does decreasing the dose of AAV and altering the route of administration help overcome the issue of neutralizing antibodies (nAbs)? If so, please provide a detailed discussion.

Response 8: We suggest that changing the method of administration may help overcome the problem of antibody neutralization, but simply reducing the dose of AAV is not enough to achieve this. For a more detailed discussion, we have expanded sections 5.4 and 5.5, and have also moved the information on methods of introduction from chapter 3.1.1 in order to combine all the information on this topic in one section.

Round 2

Reviewer 2 Report

Comments and Suggestions for Authors

1. Page 17, Lines 588–593: The authors did not cite the original article supporting the following statement. Referring to the source using only a PMID is not appropriate in professional writing. The authors should provide the full citation to the references list and cite it properly in the text.

2. Section 5.3.2. Degradation of nAbs by pre-injection of enzymes that break down Ig:
The authors have used PMIDs in multiple places throughout the manuscript. This is not considered appropriate in formal scientific writing. Instead, the authors should remove the PMIDs and replace them with proper in-text citations that correspond to the full references listed in the reference section.

3. Table 2: PMIDs are used in place of properly formatted citations. The authors should replace PMIDs with correctly styled references consistent with the citation format used throughout the manuscript and ensure that full details are included in the reference list.

Author Response

We thank the reviewer for the valuable comments. We have improved the manuscript according to the suggestions.

Comments 1: Page 17, Lines 588–593: The authors did not cite the original article supporting the following statement. Referring to the source using only a PMID is not appropriate in professional writing. The authors should provide the full citation to the references list and cite it properly in the text. 

Response 1: We appreciate the reviewer's attention to detail and agree that the correct design of citations is essential for academic rigor. The use of PMIDs was a temporary measure during the revision stage, but we recognize that this is inappropriate in the final manuscript. All links have been replaced appropriately according to the style guidelines. And we also cited the original articles in the text.

Comments 2: Section 5.3.2. Degradation of nAbs by pre-injection of enzymes that break down Ig: The authors have used PMIDs in multiple places throughout the manuscript. This is not considered appropriate in formal scientific writing. Instead, the authors should remove the PMIDs and replace them with proper in-text citations that correspond to the full references listed in the reference section.

Response 2: We agree that proper citation formatting is essential for maintaining academic rigor. During the revision process, we used PMIDs as a temporary measure, but we understand that this practice is not appropriate for the final manuscript. We’ve replaced all links with appropriate citations according to the journal's style guide.

Comments 3: Table 2: PMIDs are used in place of properly formatted citations. The authors should replace PMIDs with correctly styled references consistent with the citation format used throughout the manuscript and ensure that full details are included in the reference list.

Response 3: All links have been replaced appropriately according to the journal's style guidelines. We have also carefully checked all references to ensure consistency and completeness of the revised manuscript.

Round 3

Reviewer 2 Report

Comments and Suggestions for Authors

The manuscript can be accepted for publication.